# Revisiting Time Series Outlier Detection: Definitions and Benchmarks

**Kwei-Herng Lai**
Rice University
khlai@rice.edu

**Daochen Zha**
Rice University
daochen.zha@rice.edu

**Junjie Xu**
Penn State University
jmx5097@psu.edu

**Yue Zhao**
Carnegie Mellon University
zhaoy@cmu.edu

**Guanchu Wang**
Rice University
hegsns@rice.edu

**Xia Hu**
Rice University
xiahu@rice.edu

## Abstract

Time series outlier detection has been extensively studied with many advanced algorithms proposed in the past decade. Despite these efforts, very few studies have investigated how we should benchmark the existing algorithms. In particular, using synthetic datasets for evaluation has become a common practice in the literature, and thus it is crucial to have a general synthetic criterion to benchmark algorithms. This is a non-trivial task because the existing synthetic methods are very different in different applications and the outlier definitions are often ambiguous. To bridge this gap, we propose a behavior-driven taxonomy for time series outliers and categorize outliers into point- and pattern-wise outliers with clear context definitions. Following the new taxonomy, we then present a general synthetic criterion and generate 35 synthetic datasets accordingly. We further identify 4 multivariate real-world datasets from different domains and benchmark 9 algorithms on the synthetic and the real-world datasets. Surprisingly, we observe that some classical algorithms could outperform many recent deep learning approaches. The datasets, pre-processing and synthetic scripts, and the algorithm implementations are made publicly available at https://github.com/datamllab/tods/tree/benchmark.

## 1 Introduction

Detecting outliers from time series data has broad applications in various domains, such as manufacturers [1], edge devices [2] and HVAC systems [3, 4, 5]. Many algorithms have been proposed for time series outlier detection, including prediction-based models such as auto-regression [6] and recurrent neural networks [7], majority modeling approaches such as isolation forest [8] and autoencoder [9], and discords analysis methods such as subsequence clustering [10] and matrix profile [11].

Despite these efforts of advancing algorithm design, very few studies have investigated how we should benchmark the existing algorithms. While some real-world datasets could be used for benchmarking, they often exhibit a mixture of different types of outliers, making it challenging to understand the pros and cons of algorithms. For example, in the NYC taxi dataset [12] (left-hand side of Figure 1), the subsequence highlighted in grey is an outlier because it has significantly smaller values and forms a downhill while the majority subsequences are uphills; whereas the subsequence marked in blue is an outlier because of its wider valley. Simply obtaining an overall performance on this dataset will not help explain which types of outliers an algorithm can or cannot deal with. Moreover, labeling datasets are often laborious and expensive. In practice, as pointed out in [13], real-world datasets can be mislabeled with flaws. Thus, researchers often resort to synthetic datasets [2, 14, 15, 16, 17, 18] since they can conveniently isolate the outlier types to clearly interpret how the algorithms behave.

35th Conference on Neural Information Processing Systems (NeurIPS 2021) Track on Datasets and Benchmarks.

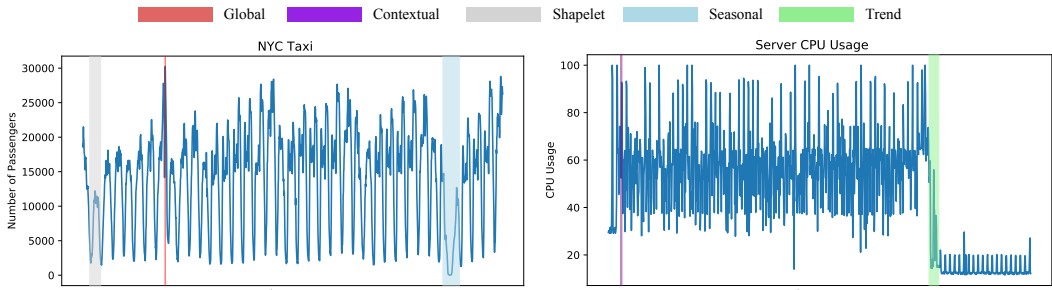

Figure 1: The three collective outliers of shapelet, seasonal, and trend in two real-world datasets [12] have very different behaviors but are regarded as the same type of outlier with the existing definition.

However, it is non-trivial to properly generate synthetic datasets for time series data due to the ambiguity of outlier definitions. The synthetic methods can be very different in the literature [2, 14, 15, 16, 17, 18] because it highly depends on how the outliers are defined. Most papers in this research line simply follow and extend the outlier definitions in non-sequential data and categorize them into point, contextual, and collective outliers [19, 20, 21, 22], illustrated in Figure 3. Unfortunately, this categorization often still relies on the similarity among points and does not model the general temporal structures in time series data [23, 24, 25]. As such, the definitions can be unclear due to the ambiguity of contexts[1]. For example, in Figure 1, the outliers marked in grey, blue and green have very different behaviors with an unusual shape, lower seasonality, and decreasing trend, respectively. However, these three outliers will be all regarded as collective outlier under the existing taxonomy. Following this, it would be challenging to synthesize these outliers due to the ambiguity of contexts.

Some previous efforts [26, 27, 28, 29] have discussed how we should generate synthetic datasets. For example, researchers have synthesized time series data with anomalous individual points to simulate the failures or intrusions in domains such as electricity load monitoring [14], edge device faults [2] and server intrusion monitoring [15]. Meanwhile, previous work has tried injecting synthetic collections of points to a sinusoidal wave to simulate the unusual events and behaviors in applications such as power plant and ECG monitoring [16, 17, 18]. While these studies have shed light on how to synthesize data, their outlier definitions only focus on their specific applications, and the resulting synthetic outliers are often only designed for a target domain. Therefore, it remains unclear how to synthesize outliers to benchmark the algorithms since we do not have a general synthetic criterion.

To bridge this gap, we aim to take a closer look into the outlier definitions in time series data and benchmark the synthetic methods and the existing algorithms. In particular, we will investigate the following questions: 1) Can we develop a taxonomy that can better categorize the outliers (e.g., the seasonal, shapelet, and trend outliers shown in Figure 1) to guide the design of synthetic datasets? 2) How can we effectively synthesize different types of outliers to better understand the capabilities of different algorithms. Through answering these questions, we make the following contributions:

- We propose a behavior-driven taxonomy for time series outliers, illustrated in Figure 2b. It views time series data with empirical observations and spectral analysis, and categorizes outliers into point- and pattern-wise outliers accordingly with clear context definitions.
- Following the behavior-driven taxonomy, we present a general synthetic criterion based on the new definitions. We also generate 35 synthetic datasets for benchmarking.
- In addition to synthetic datasets, we identify four multivariate real-world datasets that cover both point- and pattern-wise outliers from different application domains.
- We conduct extensive experiments on the synthetic and the real-world datasets to benchmark 9 algorithms, including prediction-based models, majority modeling approaches, and discords analysis methods. We surprisingly observe that some classical algorithms could outperform many recent deep learning approaches for all types of outliers. We also interestingly observe that some algorithms are able to detect certain types of pattern-wise outliers even if they are designed for point outliers. With the hope that these insights could motivate future works, we have open-sourced all the datasets, the pre-processing and synthetic scripts, and the algorithm implementation in TODS [30].

---

[1]In this work, context generally refers to a specific pattern of the rest of the data points, and can be the values of the data points globally or in a surrounding window, or general patterns such as trend and seasonality.

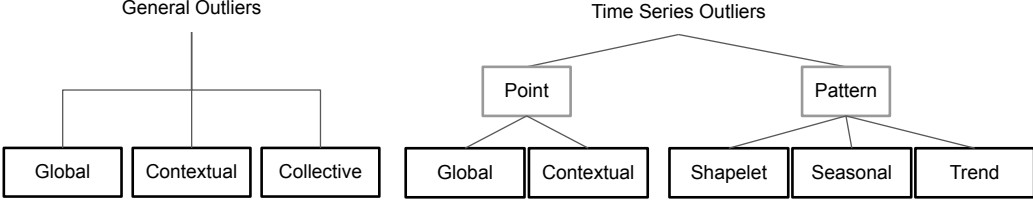

(a) Existing Taxonomy    (b) Behavior-Driven Taxonomy

Figure 2: Comparison of the behavior-driven taxonomy with the existing taxonomy. We categorize sequential outliers into point and patten-wise behaviors with clear definitions of contexts.

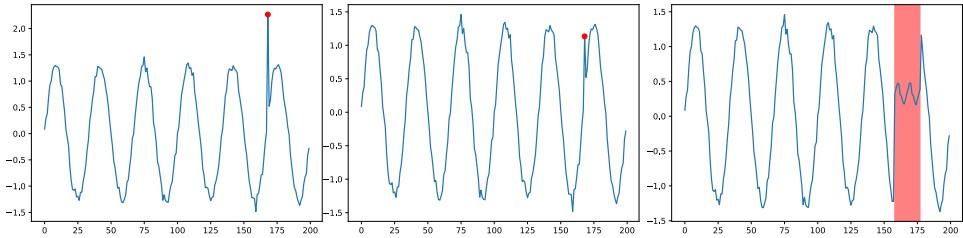

Figure 3: Examples of point (left), contextual (middle), and collective (right) outliers.

## 2 Background

This section gives a background of the previous outlier definitions in time series data. Outliers in non-sequential data are often defined as the data instances that significantly deviate from the majority of the instances [31, 32, 33, 34, 35, 36]. However, it is non-trivial to define outliers in time series data due to the temporal correlations among observations. Existing studies often follow the outlier definitions in non-sequential data. Specifically, they define the outliers in sequential data with behavior analysis [19, 20, 21, 22, 37, 38, 39, 40] and categorize them into point, contextual, and collective outliers. Figure 3 illustrates the three types of outliers that often serve as a de-facto-standard:

- **Point outlier** is defined as the individual instance that is anomalous with respect to the rest of the data. The extreme values could lead to serious consequences, and therefore point outlier is often the focus of sequential outlier detection research.
- **Contextual outlier** is the individual instance that is anomalous under a specific context, such as the discord points within the same harmonic pattern. Contextual outliers usually have relatively larger/smaller values in their own context but not globally. Identifying contextual outliers is often considered more challenging and is extensively explored in the literature [41, 42, 43, 44].
- **Collective outlier** is defined as a collection of related data instances that is anomalous with respect to the entire data set. Specifically, the individual points of a collective outlier may not be anomalous by themselves but the co-occurrence of them becomes an outlier. Collective outliers are ubiquitous in sequential data since there are often strong dependencies among time points.

Although the above categorization has covered both individual instances and collections of instances, it remains non-trivial to clearly define the collective and contextual outliers due to the ambiguity of contexts. The contexts of the contextual outliers are often very different in the literature. They can be a small window containing the neighboring points [45] or the points with similar relative positions in terms of seasonality [41]. Similarly, collective outliers can only be clearly defined with a clear context. For example, in Figure 1, the shapelet, seasonal and trend outliers have totally different behaviors under different contexts. However, the current taxonomy will categorize all of them as collective outliers since they are all outliers for multiple time points. To bridge this gap, this work aims to refine the sequential outlier definitions with clear and unified definitions of the contexts.

## 3 Revisiting Outlier Definition and Synthesizing Criteria

This section introduces a new taxonomy for time series outliers. We first revisit and motivate the behaviors of time series data with empirical observations and spectral analysis. Then we propose a new taxonomy for point- and pattern-wise outliers with clear context definitions. Finally, we discuss the existing synthetic methods and present a general synthetic criterion based on our new definitions.

### 3.1 Behaviors in Sequential Data

The most common way to model time series data is based on empirical observation [46], which treats the data as a series of data points and studies the relationships among the points. Formally, a time series data $X$ with $t$ timestamps can be represented as an ordered sequence of data points:

$$X = (x_1, x_2, \cdots, x_t), \tag{1}$$

where $x_i$ is the data point at timestamp $i$ ($i \in T$, where $T = \{1, 2, ..., t\}$). This formulation can be naturally extended to a multivariate counterpart by adding a dimension to $x_i$. Previous work often follows empirical observation to study point, contextual, or collective outliers [19, 20, 21, 22]. However, such formulation does not consider the temporal structure of the data such as trend and seasonal information. For example, given two anomalous subsequences with unusual shapelet and abnormally high frequency respectively, they will be both identified as collective outliers. This makes it difficult to analyze the cause of outliers and understand the performance of detection algorithms.

To better model the temporal structure of the time series data, we can alternatively view the data with spectral analysis [23]. The most common way of spectral analysis is to formulate the time series data as a combination of sinusoidal wave [24]: $X = \sum_n A sin(2\pi\omega_n T) + B cos(2\pi\omega_n T)$, where $sin(2\pi\omega_n T)$ and $cos(2\pi\omega_n T)$ are shapelet functions that transform a series of timestamps $T = \{1, 2, ..., t\}$ into values, and $A$ and $B$ are coefficients to define the value range. $X$ is obtained by summing up the values of multiple waves with different frequencies, and $\omega_n$ denotes the frequency of wave $n$. Although the sinusoidal wave can well represent the shapelets and seasonality of the data, it can not model trend. To tackle this issue, we adopt structural modeling [23, 25, 24] with spectral analysis to represent the time series as the combination of trend, seasonality and shapelets:

$$X = \rho(2\pi\omega T) + \tau(T), \tag{2}$$

where $\rho(2\pi T, \omega) = \sum_n [A sin(2\pi\omega_n T) + B cos(2\pi\omega_n T)]$ is the base shapelet function to approximate de-trend series (here, $\omega = \{\omega_1, \omega_2, .., \omega_n\}$ for brevity), and $\tau(\cdot)$ denotes a trend function that models the general direction of the series. This formulation can represent various shapelet patterns, such as sawtooth wave and square wave, with various trends. For example, for square sine wave with linearly increasing trend, we can set $A = \frac{1}{2n+1}$, $B = 0$, $\omega_n = 2n + 1$ ($n \in \{0, 1, ..., N\}$), and $\tau$ as a linear function $\tau(T) = T$, where $N$ controls the level of squareness.

### 3.2 Refining Sequential Outlier Definitions

The existing taxonomy for time series data mainly focuses on individual data points, e.g., point and contextual outliers. While collective outlier considers subsequences, it simply regards a subsequence as a combinatorial behavior of multiple points, which ignores the spectral information of subsequences. In this subsection, we propose a new taxonomy, shown in Figure 2b. We refine the outlier definitions in time series and identify five types of outliers that cover point- and pattern-wise behaviors.

#### 3.2.1 Point-wise Outliers

Point-wise outliers refer to unexpected incidents on individual time points. Anomalous behaviors of one time point can be a glitch or spike, where spike is an individual point with extreme value comparing to the rest of the points and glitch is an individual point with relatively deviated value from its neighboring points. Following this intuition, given a time series $X = (x_1, x_2 \cdots, x_t)$, two outlier types can be defined under point-wise behavior with different thresholds $\delta$:

$$|x_t - \hat{x}_t| > \delta, \tag{3}$$

where $\hat{x}_t$ is the expected value, which can be the output of a regression model, or simply the global mean value or mean value of a context window.

**Global outliers** refer to the points that significantly deviate from the rest of the points. They are usually the spikes in the series and therefore the threshold can be defined as

$$\delta = \lambda \cdot \sigma(X), \tag{4}$$

where $\sigma(X)$ is the standard deviation of the time series and $\lambda$ controls the range.

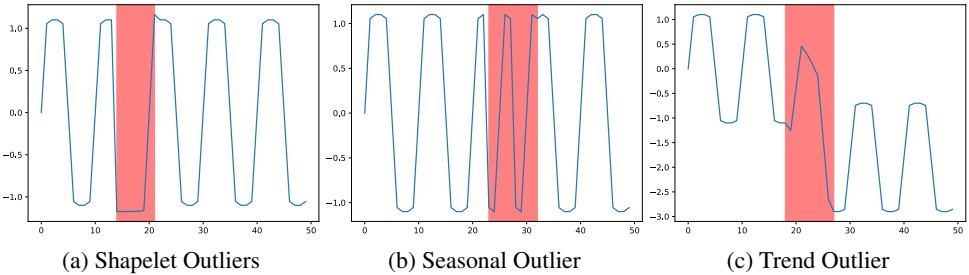

(a) Shapelet Outliers     (b) Seasonal Outlier     (c) Trend Outlier

Figure 4: Illustration of three types of pattern outliers.

**Contextual outliers** are the points that deviate from its corresponding context, which is defined as the neighboring time points within certain ranges. This type of outlier are the small glitches in the sequential data and can be defined as:

$$\delta = \lambda \cdot \sigma(X_{t-k,t+k}), \tag{5}$$

where $X_{t-k,t+k} = (x_{t-k}, x_{t-k+1}, x_{t-k+2} \cdots, x_{t+k})$ refers to the context of the data point $x_t$ with a context window size $k$, and $\lambda$ controls the threshold.

### 3.2.2 Pattern-wise Outliers

Pattern-wise outliers are anomalous subsequences, which are typically discords or inharmonies. There are three major causes of pattern-wise outliers: basic shapelet, seasonality changes and trend alternations. Specifically, given a time series data $X$, an underlying subsequence $X_{i,j}$ starting from timestamp $i$ to $j$ can be represented by a shapelet function with trend and seasonality:

$$X_{i,j} = \rho(2\pi\omega T_{i,j}) + \tau(T_{i,j}), \tag{6}$$

where $\rho$ defines the basic shape of the subsequence, $\omega$ is the seasonality of the subsequence, $\tau$ is the trend function describing overall direction of $X_{i,j}$. By analyzing three components individually, we identify three types of outliers in pattern-wise behavior, illustrated in Figure 4.

**Shapelet outliers** refer to the subsequences with dissimilar basic shapelets compared with the normal shapelet, which can be defined as

$$s(\rho(.), \hat{\rho}(.)) > \delta, \tag{7}$$

where $s$ is a function measures the dissimilarity between two subsequences, such as dynamic time warping [47]. $\hat{\rho}$ is the basic shapelets of expected subsequence, and $\delta$ is a threshold.

**Seasonal outliers** are the subsequences with unusual seasonalities compared with the overall seasonality. They have similar basic shapelet and trend but with unusual seasonalities, defined as

$$s(\omega, \hat{\omega}) > \delta, \tag{8}$$

where $\hat{\omega}$ is the seasonality of expected subsequences, and $\delta$ is a threshold.

**Trend outliers** indicate the subseuqences that significantly alter the trend of the time series, leading to a permanent shift on the mean of the data. This type of outlier retains basic shapelet and seasonality of the normalities but the slope of the trend changes drastically, which can be defined as:

$$s(\tau(.), \hat{\tau}(.)) > \delta, \tag{9}$$

where $\hat{\tau}$ is the trend of normal subsequences, and $\delta$ is a threshold.

### 3.3 Synthesizing Outliers

Introducing synthetic outliers into anomaly-free data is a very common strategy to evaluate detection algorithms. One of the synthesizing strategy is to inject sporadic outliers in an additive manner [14, 15]. Specifically, outliers are synthesized by adding the original data point with mean and standard deviation of the whole data to ensure their outlierness. Another strategy [16, 17, 18] is to replace the existing subsequences with dishormonic patterns, e.g., randomly replacing a subsequence of a cosine wave with a sinusoidal wave. Table 1 compares the synthetic methods adopted from different

| Attributes \ Domain | Edge Device [2] | Electric Load [14] | Server Log[15] | Power Plants [16, 17] | ECG [18] | SEQ (ours) |
|---|---|---|---|---|---|---|
| Point Behavior | ✓ | ✓ | ✓ | ✗ | ✗ | ✓ |
| Pattern Behavior | ✗ | ✗ | ✗ | ✓ | ✓ | ✓ |
| Point Global | ✓ | ✗ | ✓ | ✗ | ✗ | ✓ |
| Point Contextual | ✗ | ✓ | ✓ | ✗ | ✗ | ✓ |
| Pattern Shapelet | ✗ | ✗ | ✗ | ✓ | ✓ | ✓ |
| Pattern Seasonality | ✗ | ✗ | ✗ | ✗ | ✓ | ✓ |
| Pattern Trend | ✗ | ✗ | ✗ | ✗ | ✗ | ✓ |

Table 1: Comparison to synthetic methodologies in existing works from different domains.

applications. While these studies have introduced various synthetic strategies, they only focus on their specific applications, and can not serve as a general synthetic criterion for benchmarking. Moreover, none of them consider the trend outlier. In this subsection, we introduce a general and unified synthetic criterion to benchmark the evaluation of different types of outliers.

**Global Outlier.** Following Equation 4, we can synthesize a global outlier by letting $\hat{x}_t = \mu(X)$ and $\delta = \lambda \cdot \sigma(X)$, i.e., $x_t = \mu(X) \pm \lambda \cdot \sigma(X)$, where $\mu(X)$ denotes the mean, $\sigma(X)$ denotes the standard deviation, and $\lambda$ controls how much $x_t$ deviates from the expected value.

**Contextual Outlier.** Contextual outliers are expected to locally rather than globally deviate from the expected value. Based on Equation 5, we can set $\hat{x}_t = \mu(X_{t-k,t+k})$, $\delta = \lambda \cdot \sigma(X_{t-k,t+k})$, i.e., $x_t = \mu(X_{t-k,t+k}) \pm \lambda \cdot \sigma(X_{t,t+k})$, where $\mu$ and $\sigma$ are instead obtained from a subsequence.

**Shapelet Outlier.** Following Equation 7, we can synthesize a shapelet outlier from timestep $i$ to $j$ by setting $\rho$ to be other shapelets with $X_{i,j} = \rho(2\pi\hat{\omega}T_{i,j}) + \hat{\tau}(T_{i,j})$, where $\hat{\omega}$ denotes the expected seasonality, $\hat{\tau}$ denotes the expected trend, and $\rho$ is another shapelet. For instance, we can set $\rho$ to be square wave to synthesize a shapelet outlier in a sine wave, illustrated in Figure 4a.

**Seasonal Outlier.** Based on Equation 8, we can similarly synthesize a seasonal outlier from timestamp $i$ to $j$ with $X_{i,j} = \hat{\rho}(2\pi\omega T_{i,j}) + \hat{\tau}(T_{i,j})$, where $\omega$ is another seasonality while $\hat{\rho}$ and $\hat{\tau}$ are the expected ones. Figure 4b gives an example of seasonal outlier by setting the seasonality as $2\hat{\omega}$.

**Trend Outlier.** Similarly, we can follow Equation 9 to synthesize the trend outliers with $X_{i,j} = \hat{\rho}(2\pi\hat{\omega}T_{i,j}) + \tau(T_{i,j})$. Figure 4c shows the example with $\tau(T_{i,j}) = \{-1, -2, -3, \cdots, -(j-i+1)\}$.

**Discussion.** Unlike the existing definitions and synthetic methods, we introduced a structural time series model to describe pattern-wise behaviors for the following reasons. First, this formulation can provide clear contexts to describe the structural patterns and define the shapelet, seasonal, and trend outliers, which cannot be achieved by simply regarding a subsequence as a collection of points. Second, following this formulation, we can synthesize different types of outliers by inserting other shapelet, seasonal, or trend patterns. This enables us to isolate the outlier types and focus on a specific type, making it convenient to analyze and interpret how the existing algorithms behave.

# 4 Benchmark Experiments

In this section, we introduce 35 synthetic datasets based on the proposed criterion and identify four real-world multivariate sequential data which cover both point- and pattern-wise outliers. We further benchmark 9 existing algorithms implemented in TODS project [30] on these datasets. In what follows, we first describe the details of the synthetic datasets and the real-world datasets, and then elaborate on the included algorithms. Finally, we present the benchmark results and analysis.

## 4.1 Descriptions of the Datasets

We conduct benchmark experiments in unsupervised setting. Each of the algorithm is trained and tested on the same dataset. The outliers are identified based on the outlierness score generated by individual algorithms with a given contamination ratio. The benchmark experiments are conducted on both synthetic and real-world datasets as follows:

**Synthetic Datasets.** The goal of synthetic datasets is to examine the ability of algorithms to identify 5 type of proposed outliers. We generate 35 synthetic datasets with 20 univariate and 15 multivariate datasets to examine the existing algorithms in detail. Specifically, we adopt sinusoidal wave as the

base shapelet to generate 20 univariate sequential data with different ratio of outliers, where each dataset only include one kind of outlier. Then, we also generate 15 multivariate sequential data which combine different kinds of outliers into single dataset.

**Real-world Datasets.** We identify four public available real-world datasets from four different application scenario with two event-driven application and two time-based application: credit card fraud detection, IoT for drinking water monitoring, server attack monitoring and extreme space weather detection. The credit card transaction data [48] [2] and server monitoring data [49] [3] are event-driven sequential data, which contain point-wise outliers. The IoT data [50] [4] and space weather data [51] [5] are time series data, which contain pattern-wise outliers.

More datasets details are provided in Appendix B.

## 4.2 Sequential Outlier Detection Algorithms

Existing sequential outlier detection algorithms can be categorized into three types based on their working mechanisms: prediction deviation, majority modeling and discords analysis.

**Prediction Deviation** identifies the outliers by measuring the gaps between the predicted values and the original data. The assumption behind this type of algorithms is that the given data is reconstructable through regression analysis; if an individual instance is not regressable, then it is very likely to be an outlier. Autoregression (**AR**) [6] assume that each individual instance is linearly correlated to its past few instances. Gradient boosting regression (**GBRT**) [52] handles time series data in windowed-fashion and perform regression based on segmented subsequences. Derived from autoregression, recurrent neural networks with long short term memory units (**LSTM-RNN**) [7] is adopted to model the nonlinear temporal correlations between data instances.

**Majority Modeling** assumes that normal data instances are compact in hyperspace [53, 54]. It aims at identifying the decision boundary between outliers and normalities through modeling the regular data distribution. One-class SVM (**OCSVM**) [55, 56, 57] maximizes the margin between origin and the normalities and define the decision boundary as the hyper-plane that determines the margin. Isolation forest (**IForest**) [8, 58] builds an ensemble of binary trees to isolate the data points and defines the decision boundary as the closeness of an individual instance to the root. Autoencoder (**AE**) [9] maps the data points into low dimensional latent space, reconstructs the data points from the latent space representations, and defines the decision criteria by assuming the reconstruction error of outliers are significantly larger than normalies. Generative adversarial nerwork (**GAN**) [59] performs min-max optimization with a generator and a discriminator, where discriminator aims at modeling the normalities and generator targets on generating outliers that can be identified as normalities by discriminator. The decision criterion is defined as the discriminator loss on individual instances.

**Discords Analysis** measures the similarity [60] between subsequences and aims at identifying discords as outliers. Specifically, sequential data will be segmented into subsequences by a sliding window. Then, different distance computation will be performed to evaluate the discordance of each subsequence. Discords analysis is usually adopted to identify pattern-wise outliers. Subsequence clustering [10] leverages unsupervised algorithms such as OCSVM [57] and IForest [58] with segmented subsequences to detect pattern-wise outliers. Matrix profile (**MP**) [11, 61] constructs distance profiles by computing minimum distances of each subsequence to the rest of subsequences, then identifies anomalous subsequence based on the distance profile. In the benchmark, we adopt subsequence clustering with OCSVM (▲**OCSVM**) and IForest (▲**IForest**)).

For synthetic datasets, we align the contamination of all algorithms with anomaly ratio of individual dataset. As for real-world dataset, we establish 6 contamination ratio $0.01, 0.05, 0.1, 0.15, 0.2, 0.25$ and report the best result for each algorithm. More details about hyperparameters of individual algorithms are provided in Appendix C.

## 4.3 Results and Analysis

---

[2] https://www.openml.org/d/1597
[3] https://www.unb.ca/cic/datasets/ids-2017.html
[4] https://bit.ly/3fOeRvI
[5] https://dataverse.harvard.edu/dataset.xhtml?persistentId=doi:10.7910/DVN/EBCFKM

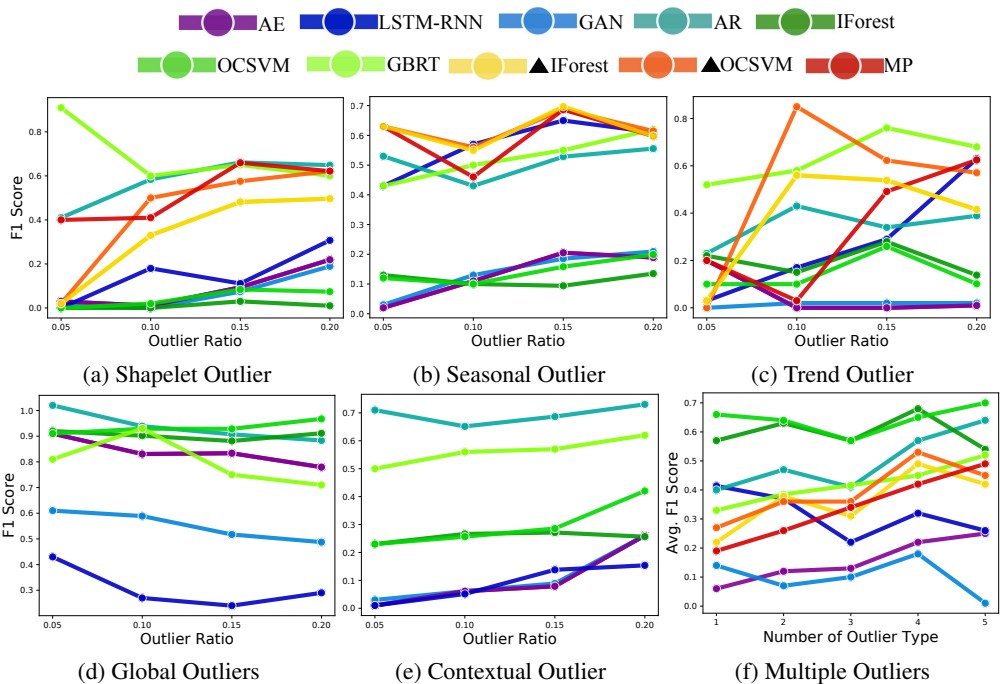

(a) Shapelet Outlier      (b) Seasonal Outlier      (c) Trend Outlier

(d) Global Outliers      (e) Contextual Outlier      (f) Multiple Outliers

Figure 5: Summary of benchmark results on univariate ($a$-$e$) and multivariate ($f$) synthetic datasets. ▲IForest and ▲OCSVM are the subsequence clustering with the two algorithms. Figure $a$-$e$ report the F1 score with respect to different ratio of outliers within the dataset and figure $f$ report the F1 score with different number of outlier types within the data. We report only prediction deviation and majority modeling-based algorithms for the point outliers. More details are provided in Appendix D.

We report the F1 score on the datasets with different outlier ratios or the numbers of involved outlier types in Figure 5 and tabulate the results of real-world datasets in Table 2. Due to the space limitation, the detailed benchmark results of synthetic datasets are tabulated in Appendix D.

**Synthetic Datasets.** Figure 5 summarizes the benchmark result on 35 synthetic datasets with F1 score. Specifically Figure 5a to 5e concludes the F1 score with respect to outlier ratio on 20 univariate sequential data and figure 5f shows the average F1 score with respect to number of involved outlier type on 15 multivariate synthetic datasets. We make the following observations.

First, classical algorithms generally outperform deep learning based methods on all of the synthetic datasets. Specifically, AR outperforms all other algorithms in detecting contextual and shapelet outliers; OCSVM and IForest outperform the rests in global outliers and multiple outliers on multivariate setting; and discord analysis algorithms perform the best in seasonal and trend outlier tasks.

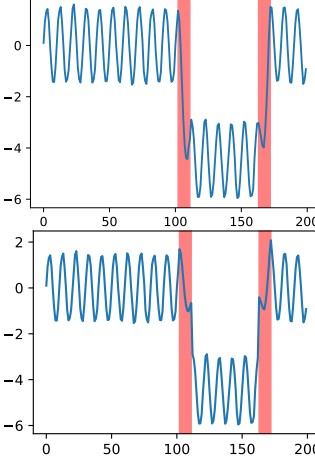

Figure 6: Before (upper) and after (lower) applying local z-normalization.

Second, detecting contextual outliers is challenging for most of the algorithms. Among all of the algorithms, only AR is able to achieve good performance. A possible reason is AR adopts contextual points to perform self regression and modeling the normalies in the context window, which benefits detecting contextual outlier.

Third, prediction-based algorithms which are designed to detect point-wise outliers are also applicable to some of the pattern-wise outliers. For example, AR outperforms all of other algorithms when detecting shapelet outlier. The reason behind this is that we adopt square sine as the anomalous shapelet to increase the difficulty. However, since the seasonality and trend of shapelet outliers

| Dataset (Best) | Credit Card | | | CICIDS | | | GECCO | | | SWAN-SF | | |
|---|---|---|---|---|---|---|---|---|---|---|---|---|
| Metrics | Precision | Recall | F1 | Precision | Recall | F1 | Precision | Recall | F1 | Precision | Recall | F1 |
| AR | 0.113 | 0.652 | 0.192 | 0.016 | 0.310 | 0.030 | 0.392 | 0.314 | 0.349 | 0.421 | 0.354 | 0.385 |
| GBRT | **0.113** | **0.657** | **0.193** | 0.018 | 0.351 | 0.034 | 0.175 | 0.140 | 0.156 | 0.447 | 0.375 | 0.408 |
| LSTM-RNN | 0.004 | 0.110 | 0.007 | **0.024** | **0.383** | **0.046** | 0.343 | 0.275 | 0.305 | 0.527 | 0.221 | 0.312 |
| IForest | 0.098 | 0.569 | 0.168 | 0.010 | 0.040 | 0.016 | **0.439** | 0.353 | **0.391** | **0.569** | **0.598** | **0.583** |
| OCSVM | 0.107 | 0.620 | 0.183 | 0.004 | 0.046 | 0.007 | 0.185 | **0.743** | 0.296 | 0.474 | 0.498 | 0.485 |
| AutoEncoder | 0.103 | 0.598 | 0.176 | 0.011 | 0.042 | 0.017 | 0.424 | 0.340 | 0.377 | 0.497 | 0.522 | 0.509 |
| ▲IForest | 0.039 | 0.226 | 0.066 | 0.011 | 0.168 | 0.020 | 0.392 | 0.315 | 0.390 | 0.406 | 0.425 | 0.416 |
| ▲OCSVM | 0.002 | 0.305 | 0.004 | 0.000 | 0.000 | 0.000 | 0.021 | 0.341 | 0.040 | 0.193 | 0.001 | 0.001 |
| MatrixProfile | 0.006 | 0.514 | 0.012 | 0.007 | 0.080 | 0.013 | 0.046 | 0.185 | 0.074 | 0.167 | 0.175 | 0.171 |

Table 2: Benchmark results on four real-world multivariate sequential data. ▲ represents the subsequence clustering based the algorithm.

remain identical to normalies, the right angle part of the synthetic outlier will be deemed as contextual outliers by AR and therefore yield an superior performance.

Fourth, local z-normalization adopted by MP may damage the performance for identifying trend outliers with different directions on zero-centered sequence when the window size is not properly set. As shown in Figure 6, with the window size that smaller than the range of outlier, the value range of the trend outlier will be similar to normal subsequences after applying the local z-normalization on the two trend outliers. Moreover, the original trend shift of the two outliers are transferred to their neighboring points, which make it hard for MP to identify the true trend outlier.

Lastly, deep learning methods such as RNN and GAN can only handle limited type of outliers. In the Figure 5f, the average F1 score of GAN and RNN tend to decrease when more types of outliers are involved. This suggests that the two algorithms might have limited performance on real-world datasets with numerous kinds of outliers or mixed type of outliers.

**Real-world Datasets.** Table 2 tabulates the best result for each algorithm on the real-world datasets. In the real-world experiments, we search the contamination ratio for all of the algorithms in {0.01, 0.05, 0.1, 0.15, 0.2, 0.25} and select the best precision, recall and F1-score to report for each dataset. Since GAN cannot identify any outliers from all of the four real-world datasets, we exclude the algorithm in the benchmark result. Based on the Table 2 we can make two observations as follows.

First, classical algorithms generally outperform deep learning methods. Except for the web attack dataset, all of other datasets are dominated by AR, IForest and OCSVM. Although this is reflected in the synthetic benchmark, it is surprising that GAN cannot identify any of the outliers within the four datasets. A possible explanation is that the outliers in real-world datasets are very complex with very different patterns, which is aligned with the result in multivariate synthetic benchmark in Figure 5f that GAN may not be able to detect outliers from dataset with numerous kinds of outliers.

Second, subsequence clustering algorithms are not robust to real-world data when combined with OCSVM. As shown in the table ▲OCSVM has the worst performance among all of the datasets with a huge gap to other algorithms. This is because the OCSVM assumes that all of the normal subsequences can be mapped into the same cluster in hyperspace, which may be not true in real-world datasets. Specifically, we observe that OCSVM with subsequence segmentation costs more than ten times of training time compared with vanilla OCSVM. This suggests that it is very challenging to find a hyperspace to cluster all normal subsequences into one class and therefore the training iteration will never stop if no maximum is set.

## 5 Discussion

As mentioned in [13], real-world outliers are complex and may not be well-labeled. This is caused by the unclear definition of the existing taxonomy, and may lead to confusion of the ability of algorithms. To better study algorithms, one approach could be creating realistic synthetic dataset with synthetic outliers, which is proposed in [13]. However, validating in real-world datasets could be preferred by researchers. To achieve this, one may leverage the proposed taxonomy on existing datasets to re-label the real-world data directly. For example, in the Taxi and the CPU datasets shown in Figure 1, the original labels are on individual points with ambiguous meanings. To address the problem, one may take a closer look to the original labeled outliers and adopt our synthetic criteria to each of the outliers

to identify the context/range of the outlier. Then, we can re-label the outliers based on the identified range/context of individual outliers towards clearer labels. Furthermore, data annotators may also refer to the proposed taxonomy and criteria to refine the labels before publishing the datasets.

# 6 Conclusion

In this work, we revisit the outlier definition in sequential data and propose a behavior-driven taxonomy to categorize time series outliers. The clear context definitions in the point- and pattern-wise behaviors make the proposed taxonomy ideal for synthesizing outliers. Based on the taxonomy, we present a general synthetic criterion with 35 corresponding synthetic datasets and identify 4 multivariate real-world datasets from different domains. We then benchmark 9 algorithms using these datasets and empirically show that classical algorithms are generally and surprisingly be superior in both synthetic and real-world datasets. We hope this insight gleaned from our benchmark experiments could motivate future algorithm design. To facilitate the reproducibility and fast experimental pipeline in time series outlier detection, we have made all the datasets, scripts, and algorithm implementations publicly available, and we will actively maintain this project. In the future, we will enrich our benchmark with more datasets and polish the definition of outliers with more delicate synthetic criteria. We will also benchmark more state-of-the-art algorithms and leverage this platform to design more effective algorithms to tackle different types of outliers.

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
