# OpenReview forum: "Revisiting Time Series Outlier Detection: Definitions and Benchmarks"
_NeurIPS.cc/2021/Track/Datasets_and_Benchmarks/Round1 — NeurIPS 2021 Datasets and Benchmarks Track (Round 1)_

### Official Review · Reviewer_kA6r · 2021-06-26
**Benchmarking time series outlier detection using an enhanced taxonomy of outliers for synthetic data generation**

**Rating:** 7
**Confidence:** 4

**Strengths:**

* A better categorization of outliers in time series data is proposed. Unlike the existing taxonomy, the behavior-driven taxonomy proposed is hierarchical with the first level based on point-wise and pattern-wise behaviors. I like the clear description in Section 3.2.2 regarding pattern-wise outliers.

* The datasets and the code for preprocessing and synthetic data generation are made publicly available.


**Weaknesses:**

* In the synthetic datasets, it is assumed that there exists only one type of outliers in each time series (or in each dimension of a multivariate time series). While this simplified setting makes it easier to assess the outlier detection capability of an algorithm for each type of outliers, real-world applications may involve multiple types of outliers even within a single time series.

* Since the pattern-wise outliers are defined based on spectral analysis, it is implicitly assumed that the time steps are evenly spaced. Unfortunately this assumption is not always true in practical applications. In fact, this assumption may not be true even for some of the real-world datasets used in the paper.


**Additional Feedback:**

* There is inconsistency between the \rho function in Eq. (2) and that in the sentence following it. The latter seems to be more suitable.

* On page 6 before Section 4, the paragraph on contextual outlier should refer to Equation 5, not Equation 4. Also, in the shapelet outlier paragraph, it is $\hat{\omega}$, not $\hat{\tau}$, that denotes the expected seasonality.


**Clarity:**

The paper is generally written clearly. There are some minor language errors that can be fixed easily if the paper is proofread carefully.


**Correctness:**

It seems that the authors regard anomalies and outliers as being the same. I am not sure if this is the common view shared by researchers working on time series data. For example, while a significant change in the trend is considered an anomaly, is it also common to call it an outlier?


**Documentation:**

The datasets and code are made available in a GitHub project. Sufficient details are provided for others to replicable the experiments reported in the paper.


**Ethics:**

The publicly available real-world datasets are provided by others. For the synthetic datasets, I do not see any ethical concerns.


**Relation To Prior Work:**

Sections 2 and 3 compare the existing taxonomy of time series outliers with the proposed behavior-driven taxonomy, and Table 1 compares the outlier types in some previous studies with those in this work. The need for this work is well justified in the deliberation.


**Summary And Contributions:**

This work seeks to benchmark existing algorithms for time series outlier detection. This is an area that has not been studied much in the research community. To facilitate the generation of suitable synthetic datasets in which different outlier types can be isolated, the paper first proposes, based on structural modeling with spectral analysis, a behavior-driven taxonomy for time series outliers involving two types of point-wise outliers and three types of pattern-wise outliers, as an attempt to unify the definitions of different outlier types. For the benchmarking study involving nine time series outlier detection algorithms ranging from conventional to recent deep learning methods, 35 synthetic univariate or multivariate time series datasets covering different outlier types according to the proposed taxonomy are used in addition to four publicly available multivariate real-world datasets.

---

> ### Author Response · Authors · 2021-07-12
> **Thank you for your feedback and suggestions. New synthetic datasets with benchmark result is available now.**
>
> **In the synthetic datasets, it is assumed that there exists only one type of outliers in each time series (or in each dimension of a multivariate time series). While this simplified setting makes it easier to assess the outlier detection capability of an algorithm for each type of outlier, real-world applications may involve multiple types of outliers even within a single time series.**
>
>  As you mentioned, we design the synthetic datasets to narrow down the problem of time series outlier detection by outlier categorization. Currently, we involve only one type of outlier in each dimension for analysis purposes. We do agree that multiple types of outliers will occur in real-world applications (as shown in Figure 1) and we have added 10 more univariate synthetic datasets regarding this situation with the benchmark result in our GitHub and provide the experimental result in the updated Appendix D (Table 11, 12, 13).
>
> **Since the pattern-wise outliers are defined based on spectral analysis, it is implicitly assumed that the time steps are evenly spaced. Unfortunately this assumption is not always true in practical applications. In fact, this assumption may not be true even for some of the real-world datasets used in the paper.**
>
> Yes, you are correct. We agree that pattern-wise outlier definition based on spectral analysis may implicitly assume that the time steps are evenly spaced and real-world datasets are way more complex for this situation. Still, we would like to mention that the two selected time-based applications are evenly-spaced. In addition, multiple pre-processing techniques are usually adopted to address the uneven spaced problem before detecting the outliers such as time-interval transformation, data imputation, moving average smoothing. As for event-driven time series such as credit card datasets, outliers are usually happening on individual transactions, making them the type of point outliers.
>
>
> **I am not sure if this is the common view shared by researchers working on time series data. For example, while a significant change in the trend is considered an anomaly, is it also common to call it an outlier?**
>
> Based on the Figure 1.2 in Outlier Analysis [1], outlier is a broader definition which covers both significant noises and anomalies. Therefore, significant change in the trend can also be considered as a type of outlier. It is also noted that outliers and anomalies are often used interchangeably in data mining literatures [2,3].
>
> [1] Charu C Aggarwal. Outlier Analysis. 2016. Springer
>
> [2] Zhao, Y., Nasrullah, Z. and Li, Z., 2019. PyOD: A Python Toolbox for Scalable Outlier Detection. Journal of Machine Learning Research, 20, pp.1-7.
>
> [3] Li, Z., Zhao, Y., Botta, N., Ionescu, C. and Hu, X., 2020, November. COPOD: copula-based outlier detection. In 2020 IEEE International Conference on Data Mining (ICDM) (pp. 1118-1123). IEEE.

---

> > ### Comment · Reviewer_kA6r · 2021-07-14
> > **Thanks for updating the paper**
> >
> > I thank the authors for addressing my comments by extending the paper which includes adding new synthetic datasets in the benchmarking study.

---

### Official Review · Reviewer_zDX2 · 2021-07-03
**An interesting work to benchmark time series outlier detection, it would help to add more analyses and details.**

**Rating:** 6
**Confidence:** 4

**Strengths:**

1. This paper identifies and tries to tackle an important problem in time series outlier detection - taxonomy to categorize time series outliers and propose general synthetic datasets for benchmarking. The proposed taxonomy and synthetic data generator are simple and effective.

2. The comparison of classic and deep outlier detection algorithms in the proposed benchmark is interesting, it motivates the future algorithm design (especially for deep outlier detection methods).

**Weaknesses:**

1. This paper proposes to generate multivariate time series data by combining different univariate synthetic time series. This is a little bit simplistic, in real-world multivariate data different signals are usually correlated to each other.

2. For the experiments on synthetic datasets, more advanced deep outlier detection algorithms are encouraged to compare. The current experimental setting seems to be unfavored for GAN-based models since most GAN-based models are trained on a training set with normal data and then tested on a noisy test dataset.


**Additional Feedback:**

The empirical evidence seems not sufficient to claim that classic methods outperform deep outlier detection algorithms. Experiments with more advanced deep learning approaches are recommended. I'd also like to see the empirical results from different experimental settings: for example, training the model on an outlier-free dataset and then testing it on a noisy test set.


**Clarity:**

The paper reads well and the text is fluent. However, figure 5 is not clear to me. Why are there different numbers of lines for different types of outliers? It would also be helpful to include more details about the synthetic experimental setting (e.g., how the training/test sets are constructed and how to select the best results for each method).

**Correctness:**

The proposed taxonomy and constructed synthetic datasets are sound. The empirical experimental analysis needs some improvement.


**Documentation:**

Yes, the authors have provided sufficient information for generating the synthetic datasets proposed in the paper.

**Ethics:**

No ethics issues.

**Relation To Prior Work:**

The background section clearly discussed the difference between the proposed work and previous works.


**Summary And Contributions:**

Motivated by the lack of unified benchmarks for time series outlier detection tasks. This paper proposes a behavior-driven taxonomy for time series outliers and presents several synthetic datasets based on their taxonomy. Empirical results show that classic outlier detection algorithms seem to be superior to deep outlier detection methods in both synthetic and real-world datasets. All the datasets and source code are made public for reproduction.

---

> ### Author Response · Authors · 2021-07-12
> **Thank you for your previous comments. Experiment setting is provided in the updated manuscript.**
>
>
> **This paper proposes to generate multivariate time series data by combining different univariate synthetic time series. This is a little bit simplistic, in real-world multivariate data different signals are usually correlated to each other.**
>
>  We agree that our multivariate synthetic dataset is simplistic compared to real-world datasets as different signals are usually correlated to each other. Still, we would like to mention that each signal in our multivariate synthetic dataset is generated based on the sinusoidal wave with different frequencies and amplification coefficients, which makes them implicitly correlated to each other.
>
>
> **For the experiments on synthetic datasets, more advanced deep outlier detection algorithms are encouraged to compare.**
>
> We agree that involving more advanced deep outlier detection algorithms is certainly a good idea for future research. Still, we would like to mention that our LSTM-RNN is the implementation of this paper [1], which has advanced techniques for thresholding prediction deviation to identify outliers. We are also working on implementing more deep learning-based algorithms such as DeepLog [2] and Telemanom [3] into our benchmark project and will open-source the code for future research.
>
> **The current experimental setting seems to be unfavored for GAN-based models since most GAN-based models are trained on a training set with normal data and then tested on a noisy test dataset.**
>
> Training GAN on the normal data and then testing on the noisy data that contains both normal samples and outliers is an interesting and related direction, which is referred to as novelty detection. Although novelty detection is closely related to outlier detection, we may not have the knowledge of normal samples in the training stage with an unsupervised setting. To focus on outlier detection and conduct fair benchmarking, all the algorithms are trained and tested under the same setting. It is certainly a meaningful direction to try novelty detection settings, and we would be happy to extend our benchmark in the future.
>
> **Figure 5 is not clear to me. Why are there different numbers of lines for different types of outliers?**
>
> For the Figure 5.a to 5.e, we provide the benchmark results on univariate synthetic datasets which involve only one type of outlier in each time series. The goal of these experiments is to examine the ability of algorithms to identify certain kinds of outliers. Figure 5.f provides the benchmark result on the situation when multiple outliers are included in one dataset, and the goal of this experiment is to examine the ability of each algorithm to handle multiple outliers at once. More train/test details can be found in the updated manuscript in Section 4.1.
>
> [1] Bontemps, Loic, James McDermott, and Nhien-An Le-Khac. "Collective anomaly detection based on long short-term memory recurrent neural networks." International Conference on Future Data and Security Engineering. Springer, Cham, 2016.
>
> [2] Du, Min, et al. "Deeplog: Anomaly detection and diagnosis from system logs through deep learning." Proceedings of the 2017 ACM SIGSAC Conference on Computer and Communications Security. 2017.
>
> [3] Hundman, Kyle, et al. "Detecting spacecraft anomalies using lstms and nonparametric dynamic thresholding." Proceedings of the 24th ACM SIGKDD international conference on knowledge discovery & data mining. 2018.

---

> > ### Comment · Reviewer_zDX2 · 2021-07-13
> > **Reply to author rebuttal**
> >
> > Thanks for the clarification. I'm looking forward to seeing your updates on more advanced deep models and the novelty detection setting.

---

### Official Review · Reviewer_N4Er · 2021-07-04
**Well-written and useful but needs work and clarity**

**Rating:** 7
**Confidence:** 4

**Strengths:**

Anomaly detection in time series is extremely well-explored and needs a more unified approach for defining different types of anomalies as well as benchmark results on different datasets. The improved taxonomy is certainly helpful and the benchmark results are also relevant to the broader machine learning community.

--- Update after receiving author feedback ---

The updated experiments, additions, and clarifications make sense.

**Weaknesses:**

1. While the idea of benchmarking anomaly detection on time-series data is certainly useful, my primary issue is with understanding the difference between the proposed paper and existing bodies of work. For example, see the following:

a. https://arxiv.org/pdf/2101.02118.pdf, which benchmarks nine  datasets  for  eight  state-of-the-art  deep-learning  models  that  were  presented  at  top-level  conferences in recent years. The paper also shows how windowed GBRT can outperform such methods. As a result, the proposed paper at the least, must include windowed GBRT as a algorithm in the pool.

b. https://arxiv.org/ftp/arxiv/papers/2009/2009.13807.pdf present UCR Time Series Anomaly Datasets which can serve as a benchmark on time series anomaly detection. They also highlight issues with existing datasets.

c. There are a vast number of publicly available datasets that have been used for benchmarking anomaly detection. While I appreciate and understand the need to create an improved taxonomy (and as a result the need to create new synthetic datasets), more number of real-world datasets should be used in the paper and a larger number of approaches should be tested.

d. Most importantly, what do the authors propose in order to bridge the move from the old datasets and taxonomy to the new one? Is there an analogy or mapping to understand the labeled data in older datasets according to the new taxonomy? How should the community perceive the vast number of labeled datasets according to the new taxonomy?

2. The proposed paper needs to clearly define the context of detection for all the algorithms, maybe have a table that categorizes between supervised and unsupervised approaches.

3.  As https://arxiv.org/ftp/arxiv/papers/2009/2009.13807.pdf point out, a major challenge in understanding prior work is the number of moving pieces and hyper-parameters that such methods have. For a benchmarking paper, I think the authors need to perform a more thorough hyperparameter search. For example, the dropout ratio for lstm-rnn, the parameters for IForest, batchsize for lstm-rnn and autoencoder, parameters for OCSVM are kept fixed.

4. Typos or grammatical errors: "While these studies have shed light on how to synthetic data", "temperal structure", "even-driven".


**Additional Feedback:**

I think this is an important paper for the community. I think a more detailed comparison with existing pieces of work (in benchmarking) and a more thorough evaluation on real-world datasets can certainly improve the paper.

**Clarity:**

Barring minor typos, the paper is well-written overall. The introduction and the background section could certainly be expanded to make them more self-complete, but have probably been kept short due to the page limit.

**Correctness:**

The claims are correct and sound. The evaluation methods for benchmark results could be improved, such as a more principled hyperparameter search.

**Documentation:**

Documentation can be improved. Please see the datasheet for datasets (https://arxiv.org/abs/1803.09010) and attach answers to all relevant questions in the appendix.

**Ethics:**

Not that I can see.

**Relation To Prior Work:**

Relation to prior work on benchmarking time series anomaly detection needs to improve.

**Summary And Contributions:**

The paper provides an improved taxonomy for anomalies in time series data. It creates 35 synthetic datasets and identifies 4 real-world datasets and benchmarks 9 algorithms on them.

---

> ### Author Response · Authors · 2021-07-12
> **Thank you for the feedback. GBRT is implemented and the result is updated. (Part 1)**
>
> **As a result, the proposed paper at the least, must include windowed GBRT as an algorithm in the pool.**
>
> Thank you for the suggestions. We have included GBRT in the benchmark experiment. In section 4.1, we have added the description of GBRT. We have also updated the results in Figure 5 as well as all the tables. Most importantly, in the experiments, we indeed observe that GBRT is also superior to deep learning algorithms in most cases.
>
> **UCR Time Series Anomaly Datasets which can serve as a benchmark on time series anomaly detection. They also highlight issues with existing datasets.**
>
> UCR time series anomaly dataset synthesizes realistic pattern-wise outliers in real-world datasets which is not suitable for analyzing the algorithm due to the complexity of the synthetic outliers and its purpose is mainly for benchmarking algorithms. In our work, we focus on outlier categorization with synthetic criteria, which leads to a dataset covering both point and pattern outliers. In addition, we provide the synthetic criteria with data generators in our project, allowing researchers to generate their own data to learn the capabilities of the algorithms. To sum up, although both of our work and [1] are discussing about the benchmark and datasets for time series outlier detection, we are working on different directions:
> * We propose a new taxonomy with synthetic criteria and provide corresponding data generators and datasets to study algorithms in detail while the UCR dataset focuses on providing realistic synthetic outliers without synthetic criteria.
> * The existing work [1] points out the problems of current real-world benchmark datasets and provides a realistic synthetic dataset which focuses on pattern-wise outlier scenarios while our work identifies the flaw of the current taxonomy of time series outliers and proposes a new taxonomy with synthetic criteria and synthetic data generators.
> * We further benchmark baseline algorithms on 35 generated synthetic datasets and 4 newly identified real-world datasets.
> * In addition, we identify a potential weakness of the local z-normalization in matrix profile on detecting trend outliers, which may not be analyzed with complex synthetic outliers.
>
> **More real-world datasets should be used in the paper and a larger number of approaches should be tested.**
>
> Thank you for the feedback. As an open-source effort, we will keep maintaining the project and involve more real-world datasets and algorithms. In this work, we mainly focus on discussing the fundamental problem of time series outlier detection and formulating the new taxonomy. We have also clearly defined the criterion to generate synthetic to help researchers to dig into the detail capability of algorithms. In the future, we will certainly involve more datasets in our open-source project.
>
>
> **What do the authors propose in order to bridge the move from the old datasets and taxonomy to the new one? Is there an analogy or mapping to understand the labeled data in older datasets according to the new taxonomy? How should the community perceive the vast number of labeled datasets according to the new taxonomy?**
>
> In this work, we propose an extended outlier taxonomy that views the time series outliers in detail. While relabeling the existing dataset based on our outlier definitions may be achievable, it may not be appropriate to alter the original label information. Therefore, there are two ways to leverage our efforts:
> * Creating realistic synthetic datasets based on the old datasets with our synthetic criteria and data generator. For example, duplicate the normal data points from the original dataset, then observe and inject the most relevant synthetic outlier into the duplicated original data.
> * In the future, data annotators may follow our outlier definition to label the new real-world data accordingly so that more detailed studies can be conducted directly on real-world datasets. For example, instead of annotating the single data points as outliers, one may identify the outliers based on the proposed outlier definition and clearly define the context for labeling the outliers. This way, the range of outliers may be clearer and easier for further analysis.
>
> Still, we provide the discussion of re-labeling existing dataset in the updated manuscript in Section 5.
>
>
> [1]  Wu, Renjie, and Eamonn J. Keogh. "Current Time Series Anomaly Detection Benchmarks are Flawed and are Creating the Illusion of Progress." arXiv preprint arXiv:2009.13807 (2020).

---

> > ### Author Response · Authors · 2021-07-12
> > **Thank you for the feedback. GBRT is implemented and the result is updated. (Part 2)**
> >
> > **The proposed paper needs to clearly define the context of detection for all the algorithms, maybe have a table that categorizes between supervised and unsupervised approaches.**
> >
> > We believe there is some misunderstanding here: First, all of our algorithms are unsupervised algorithms as labels for normal and outlier instances are usually not available. Second, while some algorithms have preferred scenarios, e.g., MatrixProfile for pattern outlier, most of the existing algorithms can be adopted to all contexts; therefore, we believe it is inappropriate to provide a table to clearly mention the context of individual algorithms.
> >
> > **I think the authors need to perform a more thorough hyperparameter search.**
> >
> > Thank you for the suggestion. We are aware that deep learning algorithms are sensitive to hyper-parameters, specifically to the neural architectures. Therefore, in our benchmark experiments, we have searched a few sets of neural architectures for deep models. Specifically, we have searched for the number of hidden layers and hidden units for autoencoder and LSTM; we also searched for the number of objects for the GAN baseline. In addition, for all of the window-based classical approaches (i.e., MatrixProfile, AutoRegression, GBRT, Subsequence Clustering), we have searched the window size and selected the best one to report. In the future, we would be happy to extend our benchmark with detailed hyper-parameter studies.

---

> > > ### Comment · Reviewer_N4Er · 2021-07-13
> > > **Post Rebuttal Comment**
> > >
> > > The updated experiments, additions, and clarifications make sense to me. I have changed my score on the paper.

---

### Author Response · Authors · 2021-07-12
**Summary of our modifications and contributions**


We sincerely thank all the reviewers for the support and the insightful comments to help improve the paper. To address the concerns of the reviewers, we have updated the paper and highlighted the updated content in blue:
* We addressed all of the typo and grammatical errors.
* We added description for the experimental setting in Section 4.1
* We added a description for the newly introduced algorithm: GBRT in Section 4.2 and Appendix C.
* We added a discussion section to discuss the possible connections to the existing datasets and future data annotations.
* We implemented and conducted benchmark experiments on the GBRT algorithm, and the result has been updated in Figure 5 and all of the tables in Appendix D.
* We conducted benchmark experiments in the setting of multiple outliers in univariate time series data. The result is tabulated in Appendix D (Table 11,12,13).
* All of the implementation, original experiment results have been uploaded to our open source project and are accessible to the public.
Thanks again to all of the precious suggestions from all of the reviewers, we look forward to further discussions.

As a closing remark, we highlight the contributions of our paper as follows.
* We propose a behavior-driven taxonomy for time series outliers, which views time series data with empirical observations and spectral analysis, and categorizes outliers into point- and pattern-wise outliers accordingly with clear context definitions. The taxonomy may be adopted for creating realistic synthetic dataset based on existing real-world datasets or enhancing data annotation on the real-world dataset in the future.
* We present a general synthetic criterion based on the new definitions with 45 synthetic datasets for benchmarking. We also identify four multivariate real-world datasets that cover both point- and pattern-wise outliers from different application domains.
* We conduct extensive experiments on the synthetic and the real-world datasets to benchmark 10 algorithms, including prediction-based models, majority modeling approaches, and discords analysis methods. We have open-sourced all the datasets, the pre-processing and synthetic scripts, and the algorithm implementation (including the updated experiments) in TODS.

---

### Decision · Program_Chairs · 2021-07-26

**Decision:**

Accept

**Comment:**

This paper introduces a set of real and synthetic datasets, benchmark algorithms, and improved taxonomy for anomalies in time series data.  Reviewers appreciated the overall contribution, though challenged the work for potential over-simplicity and existing entries in the space.